# Monitoring Long-Term Spatiotemporal Changes in Iran Surface Waters Using Landsat Imagery

**Alireza Taheri Dehkordi** [1] , **Mohammad Javad Valadan Zoej** [1] , **Hani Ghasemi** [2] , **Mohsen Jafari** [3] **and Ali Mehran** [4,*]

1   Department of Photogrammetry and Remote Sensing, K. N. Toosi University of Technology, Tehran 19967-15433, Iran
2   Department of Civil Engineering, K. N. Toosi University of Technology, Tehran 19967-15433, Iran
3   Department of Civil and Environmental Engineering, Shiraz University, Shiraz 71496-84334, Iran
4   Department of Civil and Environmental Engineering, San Jose State University, San Jose, CA 95192, USA
*   Correspondence: ali.mehran@sjsu.edu

**Abstract:** Within water resources management, surface water area (SWA) variation plays a vital role in hydrological processes as well as in agriculture, environmental ecosystems, and ecological processes. The monitoring of long-term spatiotemporal SWA changes is even more critical within highly populated regions that have an arid or semi-arid climate, such as Iran. This paper examined variations in SWA in Iran from 1990 to 2021 using about 18,000 Landsat 5, 7, and 8 satellite images through the Google Earth Engine (GEE) cloud processing platform. To this end, the performance of twelve water mapping rules (WMRs) within remotely-sensed imagery was also evaluated. Our findings revealed that (1) methods which provide a higher separation (derived from transformed divergence (TD) and Jefferies–Matusita (JM) distances) between the two target classes (water and non-water) result in higher classification accuracy (overall accuracy (OA) and user accuracy (UA) of each class). (2) Near-infrared (NIR)-based WMRs are more accurate than short-wave infrared (SWIR)-based methods for arid regions. (3) The SWA in Iran has an overall downward trend (observed by linear regression (LR) and sequential Mann–Kendall (SQMK) tests). (4) Of the five major water basins, only the Persian Gulf Basin had an upward trend. (5) While temperature has trended upward, the precipitation and normalized difference vegetation index (NDVI), a measure of the country's greenness, have experienced a downward trend. (6) Precipitation showed the highest correlation with changes in SWA ($r = 0.69$). (7) Long-term changes in SWA were highly correlated ($r = 0.98$) with variations in the JRC world water map.

**Keywords:** remote sensing; Google Earth Engine; surface water area; surface water dynamics; surface water variations; water scarcity; Iran; Landsat

## 1. Introduction

Global climate change, anthropogenic activities, and rapid urbanization have caused significant changes in Earth's water resources [1–3]. Moreover, population growth has increased water demand in various agricultural and industrial sectors [4]. Hence, monitoring water resources can alleviate impending water shortages by providing insight into spatiotemporal trends for short- and long-term planning [5].

Inland surface waters (SWs) play a critical role in land-water and biogeochemical cycles, water supply, agriculture, and food production, as well as in ecological, hydrological, and environmental processes [6–8]. Since in arid and semi-arid regions (i.e., Iran), SWs are the primary source of freshwater, the accurate detection of long-term spatiotemporal surface water area (SWA) changes is of high importance. SWA and its associated changes directly impact groundwater storage, land surface temperature, and soil moisture while, at the same time, posing severe constraints on agricultural-, industrial-, and urban-related activities [9,10].

Relative to conventional field-based surveying methods, remote sensing (RS) technology provides low-cost, high-frequency, and wide-coverage satellite imagery with various spatial resolutions acquired in diverse spectral bands. RS also allows researchers to conduct rapid studies within remote, inaccessible, and large-scale study sites [11,12]. Satellite images have been successfully utilized in a variety of applications, including the estimation of water quality parameters [13,14], water resource management [15], shoreline detection [16], and flood mapping [17]. Both active and passive satellite imageries have been deployed for SW mapping, among which the advanced very high-resolution radiometer (AVHRR) [18], moderate resolution imaging spectroradiometer (MODIS) [19], Sentinel-1 [20], Sentinel-2 [21], and Landsat [7] are the most commonly used. However, Landsat is the only freely available mission with a long historical record and medium spatial resolution. It is an optical passive joint program between the National Aeronautics and Space Administration (NASA) and the United States Geological Survey (USGS), which has proved to be effective for long-term processes and applications, such as greenhouse mapping [22], landcover dynamics [23], the monitoring of impervious land [24], and land cover change detection [25].

Utilizing remotely-sensed satellite imageries for long-term and large-scale studies requires huge storage space and powerful processing tools [3]. Google Earth Engine (GEE) is a cloud-based platform that processes data online without the need to download them [26]. GEE has archived a massive catalog of earth observation data and provides Python and JavaScript application programming interfaces (APIs). GEE performs parallel processing online, which enables researchers to work on petabytes of data rapidly. Some recent studies have effectively used GEE for various applications, such as flood mapping [27], canopy phenology [28], river width extraction [29], and landcover classification [30].

SW mapping techniques from remote sensing imagery are divided into two major categories: supervised classification (SC) models and rule-based (RB) algorithms [2,31]. For SC techniques, different input features, such as image bands and spectral indices, are used to train classification models such as support vector machines (SVMs), random forests (RF) or deep neural networks (DNNs) using well spatially-distributed training samples [32]. However, providing reliable training samples requires extensive field surveys and human expertise, which makes SC techniques a costly and time-consuming challenge for large-scale and long-term studies [33,34]. Moreover, SC techniques take longer to produce final results than RB methods [35]. In contrast, RB methods do not need reference samples and are computationally efficient and easy to implement, making them useful for large-scale and long-term studies [7,31].

RB methods can be grouped into three general categories: (1) single-index (SI), (2) multi-index (MI), and (3) transformation-based (TB) approaches. Most researchers have utilized SI techniques to map SW, in which only one multi-band water extraction index is deployed. Normalized difference water index (NDWI) (also known as $NDWI_{McFeeters}$) is one of the most famous SI methods [36]. The improved version of NDWI, modified NDWI (MNDWI or $NDWI_{Xu}$), was introduced later in 2006 with the replacement of short-wave infrared (SWIR) with near-infrared (NIR) [37]. In both NDWI and MNDWI, positive values indicate water. Feyisa et al. (2014) also developed an automatic water extraction index (AWEI), which uses two modes: shaded images with dark surfaces ($AWEI_{sh}$) and shadowless images ($AWEI_{nsh}$) [38]. In [39], the $AWEI_{sh}$, with a threshold greater than zero, was used to monitor dams and lakes in New Zealand within GEE. A novel water index ($WI_{2006}$) was also proposed by the authors of [40] to map SW, which was later improved and simplified ($WI_{2015}$) in [41]. ANDWIs (Augmented NDWIs) have also been found to be successful in SW mapping [5].

In the MI category, vegetation and water indices were used simultaneously for SW detection. The simultaneous use of MNDWI, enhanced vegetation index (EVI), and normalized difference vegetation index (NDVI) was widely used as a SW-mapping technique, especially in GEE-based studies [11,31,34]. In other studies, MNDWI was replaced with the land surface water index (LSWI) and NDWI to boost the performance of the aforemen-

tioned technique [42]. In a recent study on the Yangtze River basin, the difference between $AWEI_{sh}$ and $AWEI_{nsh}$ was used simultaneously with EVI, NDVI, and MNDWI [7] to monitor SWA changes.

In TB methods, SW can be detected by determining different rules on the transformations of the image bands. These transformations project the image band's information into a new feature space. Tasseled-cap (TC) transformation is an effective technique for compressing Landsat images into fewer bands while retaining necessary information [43–46]. TC transformation results in three components: wetness, greenness, and brightness, from which the wetness component was previously used to identify SW [43–46]. It should be highlighted that TC coefficients were mainly determined by using orthogonalization techniques such as Gram–Schmidt ($Ortho_w$) or by transforming principal component (PC)-based rotated axes ($Rot_w$) [47,48]. The performance of the wetness component, derived from the two mentioned approaches ($Ortho_w$ and $Rot_w$), has not yet been compared in SW mapping. The LBV (L: general radiance level, B: visible-infrared radiation balance, V: radiance variation vector) transformation of green, red, NIR, and SWIR1 bands has also been reported to be a well-performing technique for mapping SW from Landsat data, which was first developed by Zang on optical satellite imageries in 2007 [49,50].

There are long-term global reference water maps, the most famous of which is the Joint Research Center (JRC), which has been freely available since 1984. An expert system produces JRC, and the results are collated into monthly or yearly histories [51]. This monthly/yearly period fails to capture the extent of change in water bodies, particularly in those with high seasonal fluctuations or those prone to short-term flooding and drought [52]. Moreover, the monthly/yearly periods prevent researchers and scientists from near-real-time monitoring purposes. Other scholars have also shown apparent errors in these maps [53]. With the capability of being applied to each Landsat scene without needing ground-truth data, performance evaluations of the reviewed water mapping rules (WMRs) provide useful insights which are applicable for increasing the temporal frequency of reference maps and improving their performance [53].

As mentioned above, various WMRs have been developed for SW extraction using remote sensing data. Since previous studies have reported different results, it is indispensable to evaluate their performance before the long-term spatiotemporal change analysis of SWA [6,7,54]. For example, in Nigerien Sahel, twelve SI-based methods were tested to detect SWA changes, from which NDWI performed poorly [55], while in the Poyang Lake Basin [2], NDWI achieved the highest overall accuracy.

Iran can be a challenging study area for the evaluation of different WMRs due to its diverse land cover types. Moreover, because of being located within the Earth's dry belt, SWs play a critical role in Iran's ecological environment and national sustainable development [56]. It should be noted that SWA here refers to Iran's inland surface water extent, and the results were not affected by open waters (e.g., the Caspian Sea, Oman Sea, and the Persian Gulf). Thus, the objectives of the current study were:

- To evaluate the performance of various WMRs based on the SI, MI, and TB categorization within GEE.
- To investigate long-term spatiotemporal changes in SWA in Iran over a 32-year period using Landsat 5, 7, and 8 data.
- To examine the long-term correlation between environmental variables (such as precipitation and temperature) and SWA change.

## 2. Study Area

Iran is located in the Middle East (ME), Western Asia (Figure 1a). It covers an area of around 1.7 million km$^2$ and extends between 25°–40°N and 45°–65°E, and is ranked as the second-largest country in the ME. Iran comprises six main basins, as depicted in Figure 1b [57]. Some of the basins extend beyond Iranian boundaries. Table 1 outlines the total area, the percentage inside Iran, and the mean altitude of each basin. Moreover, Iran has 31 administrative divisions (provinces) and a population of approximately 85 million,

which is almost one percent of the world's population [57]. Iran has more than 2500 km of water borders, with the Caspian Sea in the north and the Persian Gulf and Oman Sea in the south. This country is predominantly covered by mountains and deserts, with an average altitude of 1200 m above sea level. Figure 1c depicts Iran's 90 m digital elevation model (DEM), derived from the shuttle radar topographic mission (SRTM) [58]. Iran has only 0.2% of the world's freshwater resources. Precipitation is less than one-third of the world's average and varies greatly across the country [59]. Figure 1d illustrates the 32-year mean precipitation rate obtained from the famine early warning systems network (FEWS NET) and land data assimilation system (FLDAS) [60]. Most of the precipitation happens in the northern and western basins (the Caspian Sea and Persian Gulf basins), where the 32-year mean NDVI is also higher than in other regions, as shown in Figure 1e (from National Oceanic and Atmospheric Administration (NOAA) AVHRR NDVI product [61]). High mountain ranges in the north (Alborz) and the west (Zagros) block rainfall systems from reaching the central regions. This blockade of precipitation over the central regions in the long-term has resulted in arid and semi-arid environments, such as the Kavir and Lut deserts. Due to climatic changes resulting in scarcity and an uneven distribution of precipitation across the country, water resources management is vital in Iran. In recent years, the shrinkage of important lakes and rivers, such as Lake Urmia, Lake Hamoon, and Bakhtegan, has also caused severe problems, including frequent dust and salt storms, migration, and imposes extensive pressure on different sectors, such as industry, agriculture, and urban populations [59]. It should be noted that less than 10% of the Qareh-Qum (known also as Sarakhs) Basin is located inside Iran. Thus, this article does not examine the long-term SWA changes in the Qareh-Qum Basin separately. However, its corresponding SWA is considered in the total SWA for Iran.

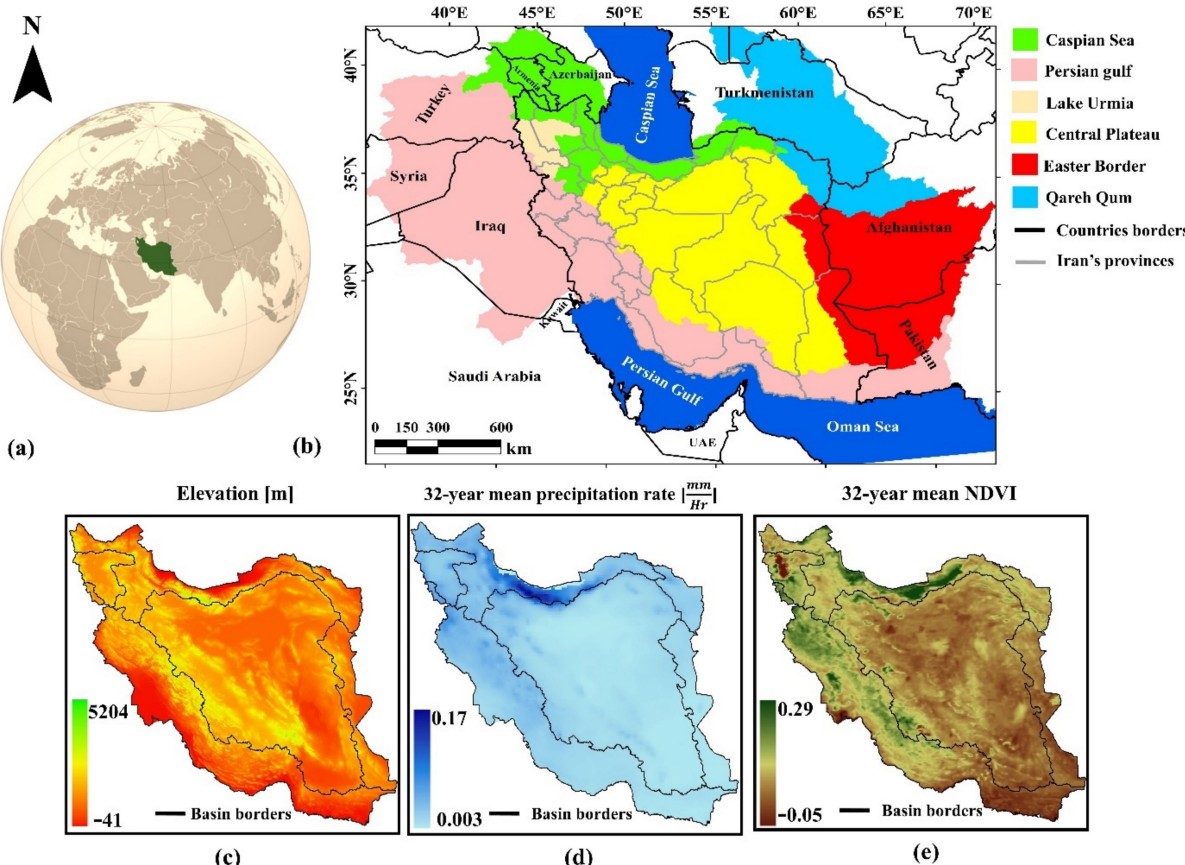

**Figure 1.** Study Area: (**a**) location of Iran, (**b**) geographical position of Iran's basins and 31 provinces, (**c**) 90 m DEM (above Geoid), (**d**) 32-year mean precipitation rate (1990–2021), and (**e**) 32-year mean NDVI (1990–2021).

**Table 1.** Iran's water basins.

| Basin | Total Area [km$^2$] | Percentage Inside Iran | Mean Altitude (Above Geoid) [m] |
|---|---|---|---|
| Caspian Sea | 346,896 | 50.5% | 1369 |
| Persian Gulf | 1,279,083 | 33.5% | 982 |
| Lake Urmia | 51,739 | 100% | 1735 |
| Central Plateau | 825,124 | 100% | 1350 |
| Easter Border [1] | 565,734 | 18.25% | 1188 |
| Qareh-Qum [2],* | 461,141 | 9.5% | 1210 |

[1] Also known as Hamoon. [2] Also known as Sarakhs. * This article does not examine the long-term SWA variations in the Qareh-Qum Basin separately (Section 4.2). However, its corresponding SWA is considered in the analysis of the total SWA.

## 3. Data and Methodology

### 3.1. Satellite Data

3.1.1. Landsat Imageries

The Landsat satellite images, which are freely accessible in GEE, were used in this study to analyze the long-term (1990–2021) variations in SWA in Iran. Landsat images had not only a medium spatial resolution for SW mapping but also a long historical record since 1972 [62]. To completely cover the 32-year period, we used images of Landsat 5 (L5) Thematic Mapper (TM), Landsat 7 (L7) Enhanced Thematic Mapper Plus (ETM+), and Landsat 8 (L8) Operational Land Imager (OLI) data [41]. At the time of conducting the research, the TC transformation coefficients were not calculated for Surface Reflectance (SR) images, so the wetness component was calculated from Top Of Atmosphere (TOA) reflectance images [46]. For the rest of the WMRs, SR images were used since they can lead to higher classification accuracies [63]. SR images were derived after Landsat Surface Reflectance Code (LaSRC) correction in the case of L8 data [63]. While Landsat Ecosystem Disturbance Adaptive Processing System (LEDAPS) correction was utilized to provide the SR of ETM+ and TM data [64]. Additionally, this study used six bands of red, green, blue, NIR, SWIR1, and SWIR2 of Landsat images [7]. An amount of 104 Landsat scenes, covering Iran completely, ranging from path 155 to 170 and row 32 to 43 (Figure 2a), were used. Due to there not being enough high-quality images available, this study only analyzed SWA variations after 1990 [11].

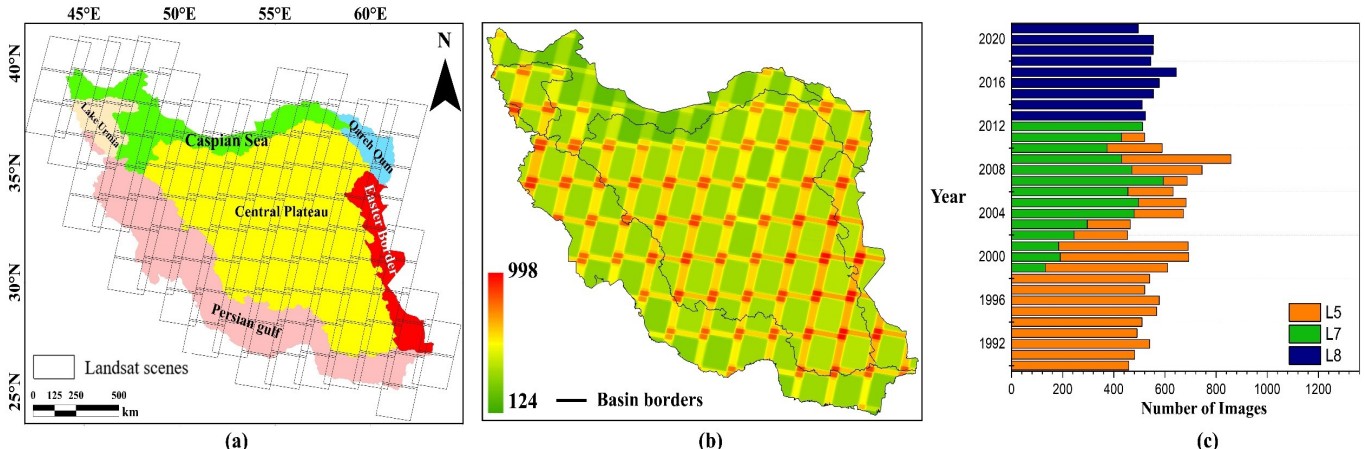

**Figure 2.** (**a**) Landsat scenes covering Iran's different basins; (**b**) number of observations per pixel during 32 years of study; (**c**) total number of images used in each year.

In Iran, July through October are the warmest months of the year. Therefore, in our framework, we only used images which were taken during those months [11,34]. Our selection of the warmest months increased the likelihood of cloudless satellite images. Moreover, July–October was the shortest time span, allowing complete image composites

to be generated over Iran every year. We only included images with 10% cloud coverage or less. In the northern parts of Iran (Caspian Sea basin), the filter on cloud coverage was increased to 15% so as to have at least one image from all parts of Iran. Figure 2b shows the total number of observations per pixel from 1990 to 2021. Also, the number of used images per Landsat mission (L8, L7, and L5) is shown in Figure 2c. This study did not use L7 images after 2013 to minimize possible scan line corrector (SLC)-related errors in the results [65]. This article used a total of 17,824 Landsat scenes over Iran to map SW.

### 3.1.2. Global Human Settlement Layer (GHSL)

The global human settlement layer (GHSL) was used to remove urban and built-up areas to increase the accuracy of SW detection [66]. The GHSL was developed using heterogeneous data with a spatial resolution of 38 m and could be accessed via GEE. In terms of spatial resolution, it was the closest free, global settlement map from the Landsat imageries [67].

### 3.1.3. SRTM DEM

The 30-m shuttle radar topographic mission (SRTM) digital elevation model (DEM) was also used to generate shadow masks to remove hill-shaded areas from all over the country [68,69]. It is the most frequently used DEM in various remote sensing applications [70], with a spatial resolution of 1 arc-second (approximately 30 m), and was gap-filled using various open-source data.

### 3.1.4. FLDAS

The famine early warning system network (FEWS NET) land data assimilation system (FLDAS) data was also used to examine the relationship between environmental variables, including precipitation and temperature, and SWA changes [60]. FLDAS was a free monthly-available product that used the Noah version 3.6.1 surface model with CHIRPS-6 hourly rainfall and was downscaled using the NASA Land Surface Data toolkit. It had a spatial resolution of about 11.1 km, which provided information on climate-related variables since 1982.

### 3.1.5. NOAA AVHRR

The mean NDVI over Iran was derived from NOAA AVHRR NDVI to explore the association between changes in greenness and SWA. It had computed NDVI values since 1981 from the NOAA AVHRR Surface Reflectance product, gridded at a resolution of 0.05° (5.5 km) [61].

### 3.1.6. USGS Spectral Library V7

The United States Geological Survey (USGS) spectral library version 7 (splib07b) was also used to analyze the spectral signatures of different land cover classes when comparing the performance of different WMRs [71]. Laboratory, field, and airborne spectrometers were used to assemble splib07b. Since there was no freely available Iranian spectral library, we used non-water samples of splib07b to vindicate the performance of WMRs.

### 3.1.7. Other Datasets

The Joint Research Center (JRC) global surface water map and quality attribute map (QA), derived from Landsat images, were also used to prepare validation data [51]. JRC was generated using about 4.5 million Landsat images with a spatial resolution of 30 m and has been available since 1984 [51]. In JRC, each pixel was classified into water and non-water based on an expert system which provided information about surface water extent and change. A water/non-water quality attribute (QA) map was also derived by applying the function of mask (Fmask) algorithm to Landsat images. QA was used in conjunction with JRC to confirm visually interpreted samples' classes (water or non-water) [72].

Additionally, two global landcover maps produced by the Environmental Systems Research Institute (ESRI) and the European Space Agency (ESA) were deployed to visually interpret the performance of different WMRs for 2020 [73,74]. The ESRI map was generated by a deep learning model trained with 5 billion hand-labeled pixels based on six bands of Sentinel-2 surface reflectance data collected from over 20,000 sites across the globe. The ESA map was also derived from 2,160,210 Sentinel-2 L2A and 291,855 Sentinel-1 ground range detected (GRD) imageries with a minimum accuracy of 75%. The ESRI and ESA maps covered 2020.

### 3.2. Framework

Our framework consisted of three general steps (Figure 3): (1) the pre-processing of remote sensing data, (2) feature extraction and SW mapping (with different WMRs), and (3) an accuracy evaluation and further analyses. The proposed framework was all developed on GEE. Each step is described in detail here.

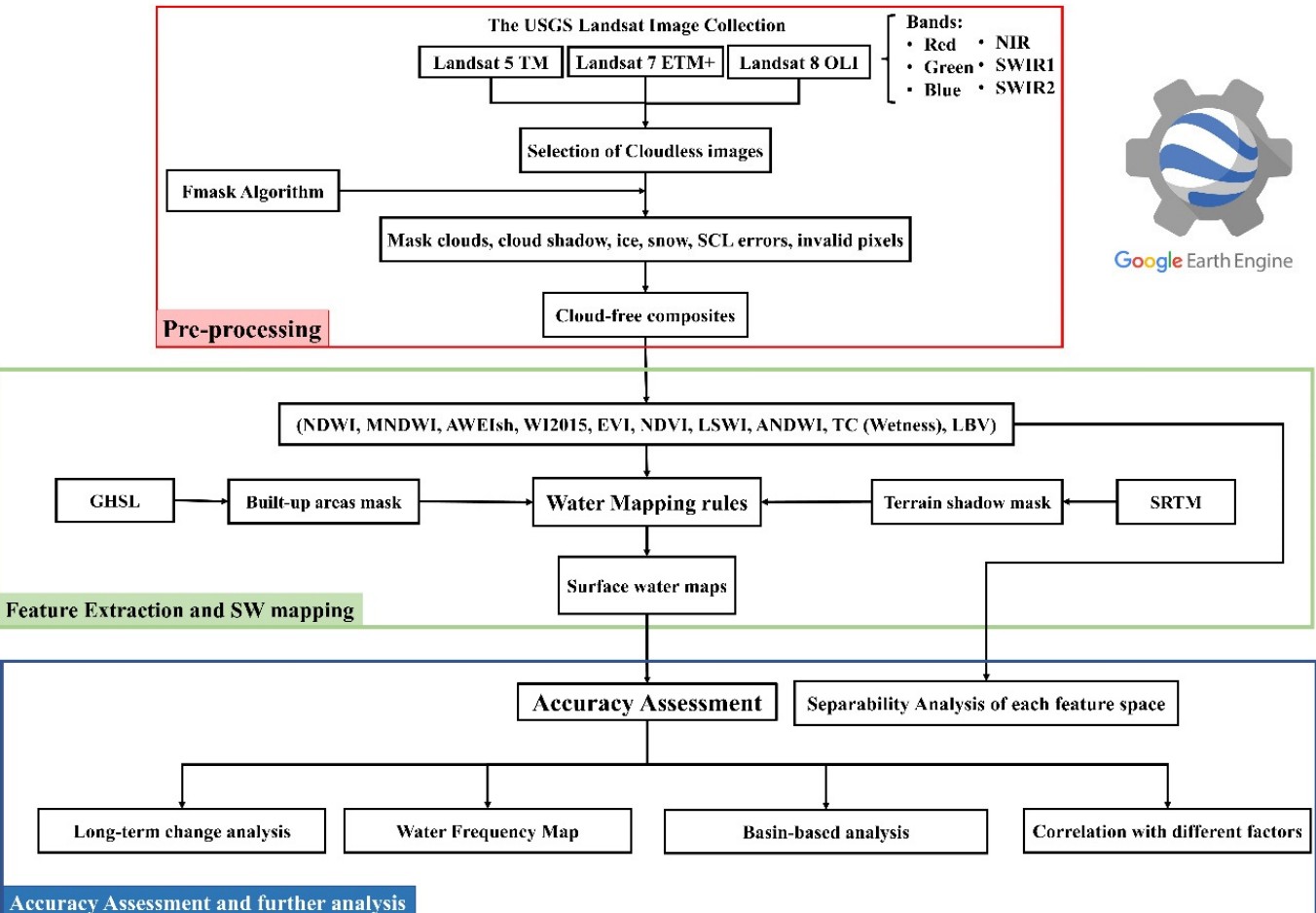

**Figure 3.** Proposed methodology framework.

#### 3.2.1. Pre-Processing

As stated in Section 3.1, our framework filtered the satellite images, and only cloudless Landsat images that passed the filtering criteria were used. The filtered images could contain invalid pixels, such as clouds, cloud shadows, snow, ice, and other sensor-related errors, such as scan line corrector (SLC) failure gaps in L7 data. To eliminate bad-quality and ineffective observations, the Fmask algorithm was applied [72]. Since many scenes covered the study area, generating a cloud-free composite for SW mapping was also necessary. To generate a cloud-free composite, a temporal-aggregation technique with a median filter was

employed, which returned the median value of all image bands at each pixel, resulting in a six-band (red, green, blue, NIR, SWIR1, and SWIR2) cloud-free composite over Iran [75]. A Median aggregating process also improved the quality of both SR and TOA images by getting rid of potential noise.

### 3.2.2. Feature Extraction and Water Mapping Rules

In this step, SRTM and GHSL data were first used to mask out the built-up and shaded areas from the composite product of the previous step. Shaded areas were removed using the command 'ee.Terrain.hillshade' in GEE, taking the SRTM DEM and azimuth angle of the sun at the time of imaging as inputs [76]. Then, different WMRs were used to map SW in the remaining regions [54]. This study aimed to compare various SW mapping methods used on the remotely-sensed images based on three categories of SI, MI, and TB. Thus, we selected 12 WMRs that were among the most common and recent SW mapping approaches (Table 2). From a total number of 12 WMRs approaches, five methods were related to the SI category (1–5), four methods were in the MI category (6–9), and the remaining three methods were TB methods (10–12). According to Table 2, the SI and MI categories used water and vegetation indices to extract water maps, whereas the TB category transformed the image bands into a new space.

**Table 2.** Water Mapping Rules (WMRs).

| Cat | Nu | Feature Space | Required Bands | Criteria | Reference |
|-----|----|---------------|----------------|----------|-----------|
| SI | 1 | NDWI | G, NIR | NDWI $> 0$ | [36] |
| | 2 | MNDWI | G, SWIR1 | MNDWI $> 0$ | [37] |
| | 3 | $AWEI_{sh}$ | G, NIR, SWIR1, SWIR2 | $AWEI_{sh} > 0$ | [38] |
| | 4 | $WI_{2015}$ | G, R, NIR, SWIR1, SWIR2 | $WI_{2015} > 0$ | [41] |
| | 5 | ANDWI | All 6 bands | ANDWI $> 0$ | [5] |
| MI | 6 | EVI, NDVI, MNDWI | B, G, R, NIR, SWIR1 | EVI $< 0.1$ and (MNDWI $>$ NDVI or MNDWI $>$ EVI) | [11,31,34] |
| | 7 | EVI, NDVI, NDWI | B, G, R, NIR | EVI $< 0.1$ and (NDWI $>$ NDVI or NDWI $>$ EVI) | [42] |
| | 8 | EVI, NDVI, LSWI | B, G, R, NIR, SWIR1 | EVI $< 0.1$ and (LSWI $>$ NDVI or LSWI $>$ EVI) | [42] |
| | 9 | $AWEI_{nsh}$, $AWEI_{sh}$, EVI, NDVI, MNDWI | All 6 bands | ($AWEI_{nsh} - AWEI_{sh} > 0.1$) and (MNDWI $>$ NDVI or MNDWI $>$ EVI) | [7] |
| TB | 10 | $Rot_w$ | All 6 bands | $Rot_w > 0$ | [47] |
| | 11 | $Ortho_w$ | All 6 bands | $Ortho_w > 0$ | [48] |
| | 12 | B, V | G, R, NIR, SWIR1 | B $>$ V | [49] |

### 3.2.3. Accuracy Assessment and Further Analysis

Selected methods (Table 2) could be used to detect SW in n-dimensional feature space (number of used features). For the SI methods, n was equal to 1; for the MI methods, n was at least 4, and for the TB methods, n was either equal to 1 or 2. First, the separation of water and non-water classes in each feature space was investigated. Transformed divergence (TD) and Jefferies–Matusita (JM) distances were two measures used to calculate the separation of the target classes (water and non-water) in each feature space [77,78]. In addition to being easy to implement, TD and JM were also the most common probabilistic- and divergence-based separation measures. JM was calculated as a function of the Bhattacharyya distance. While TD provided a covariance-weighted distance between the class means to determine whether the target classes were separable. TD and JM values ranged between 0–2, with larger values indicating better separation. With the greater separation of the target classes (water and non-water) in each feature space, the WMRs were expected to accurately identify target classes within a single threshold.

Along with examining the separability of the classes in the feature space of each method, the performance of the selected WMRs was also assessed using evaluation data, which were obtained from several field visits and the visual interpretations of high-

resolution satellite imageries. Field visits were conducted from July to October in 2020, 2019, and 2018 within different parts of Iran. Two saline regions in central parts and different lakes (Urmia, Hamoon, Tashk, and Bakhtegan Lakes) were visited. Moreover, a set of evaluation plots, each representing at least 60 m × 60 m regions (2 Landsat pixels) to ensure the purity of each sample in Landsat images, was prepared through visual interpretations of Google Earth and different band combinations from Sentinel-2 optical imageries with less than 5% cloud coverage over three consecutive years (2018, 2019, and 2020) [7,31,34,79]. To ensure the inter-class variability of the evaluation data, water samples were taken from different regions, such as reservoirs, lakes, and rivers, whereas non-water samples, were collected from agricultural lands, bare soil, grasslands, and trees. Visually interpreted samples were compared against the median composite of the JRC monthly maps from July to October and QA reference maps to have similar water and non-water classes within both [21,80,81]. A total of 5000 samples per class were provided for each year, with a uniform distribution throughout the study area to avoid evaluation biases. Additionally, ESRI and ESA landcover maps were deployed to visually interpret the performance of the different WMRs in 2020 [73,74]. The WMRs were compared based on the user accuracy (UAs) of each class and the overall accuracy (OA) of their classification [82]. OA represented the ratio of correctly classified samples in both classes of water and non-water to the total number of samples in both classes (Equation (1)). UA was the proportion of correctly classified samples within each class to the total number of samples of the same class (Equation (2)).

$$OA = \frac{C_w + C_{nw}}{N_w + N_{nw}} \tag{1}$$

$$UA_W = \frac{C_w}{N_w}, \ UA_{NW} = \frac{C_{nw}}{N_{nw}} \tag{2}$$

where $C_w$ and $C_{nw}$ are the number of correctly classified samples in water and non-water classes, respectively. Also, $N_w$ and $N_{nw}$ refer to the total number of samples in water and non-water classes.

To better compare the classification results between the different WMRs, McNemar's statistical test was used [83], indicating the significance of improvement between two classification results based on $\chi^2$ (Chi-square) distribution. Greater $\chi^2$ values revealed greater improvements in the classification results. Equation (3) gives the mathematical formula of $\chi^2$ with a confidence of 95%. Moreover, *t*-test analysis was used to statistically investigate the confidence of long-term changes in SWA and different factors, with a null hypothesis of no statistically significant relationship and an alternative hypothesis of a strong statistically significant relationship. Simple linear regression and the sequential Mann–Kendall (SQMK) tests were also used for the analysis of trends in SWA changes in Iran and different basins [84]. The SQMK test yielded two progressive and retrograde series. If they had the same trend and no intersections, then there was a significant trend in the data. Absence of a significant trend can be concluded when series intersect at several locations.

$$\chi^2 = \frac{(f_{12} - f_{21})}{(f_{12} + f_{21})} \tag{3}$$

where $f_{ij}$ is the number of samples correctly classified by classifier I and incorrectly classified by classifier j.

## 4. Results

### 4.1. Performance Evaluation of Different Water Mapping Methods

The separation of the two target classes (water and non-water) in the feature space of each WMR was analyzed in the first step of the performance evaluation for various WMRs. The separation was determined based on JM and TD measures for 2018, 2019, and 2020 (with the years having reference samples from both water and non-water classes), as shown in Figure 4a. These years were selected because the evaluation samples were related to

these years. According to the obtained results, TD values were generally higher than JM values in each feature space. Our results showed that the feature space of the 7th method from Table 2 (NDWI, EVI, and NDVI) provided the best separation between the target classes based on the TD measure, with a value of 2 for all three years of the study. The 7th method (Table 2) also showed the highest discrimination in the JM measure. However, our results suggest the $Rot_w$ feature space had the least separation capability for both of the metrics of JM and TD. NDWI and $Ortho_w$ (1st and 11th methods) were also among the best discriminative feature spaces for water and non-water classes. It should be highlighted that using vegetation indices with water indices (for example, 1st and 7th methods) resulted in higher discrimination between the target classes.

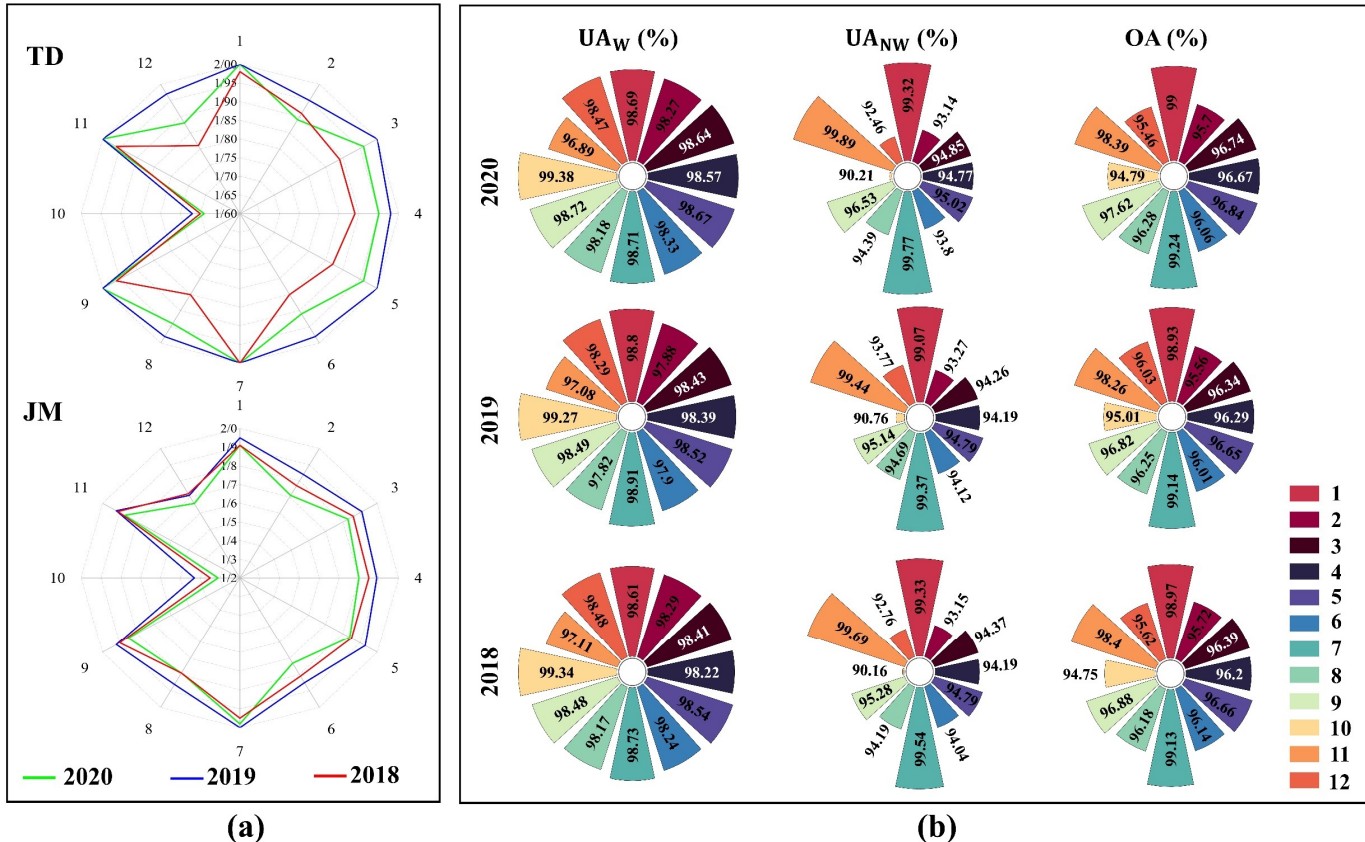

**Figure 4.** Performance analysis of 12 different WMRs over the study area: (**a**) separability analysis of target classes (water and non-water) in different feature spaces using JM and TD measures; (**b**) performance evaluation of different WMRs based on $UA_w$, $UA_{nw}$, and OA. Different methods are shown with their corresponding numbers in Table 2.

The OA, user accuracy of water class ($UA_W$), and user accuracy of the non-water class ($UA_{NW}$) for each WMR are depicted in Figure 4b. All approaches performed well in water detection, resulting in a $UA_W$ of 96.89% (11th WMR in 2020) and higher. The 1st, 7th, and 11th WMRs outperformed other approaches in terms of $UA_w$. While the highest $UA_{NW}$ was related to the 11th WMR (99.89%, 99.44%, and 99.69%), 7th WMR (99.77%, 99.37%, and 99.54%), and 1st WMR (99.32%, 99.07%, and 99.33%), respectively. All WMRs achieved an OA of at least 94.75%, among which the 7th method surpassed all other methods (with OAs of 99.24%, 99.14%, and 99.13%). In contrast, the 10th WMR ($Rot_w$) showed the poorest performance, with OAs of 94.79%, 95.01%, and 94.75% for 2020, 2019, and 2018, respectively. To sum up, in terms of $UA_W$, $UA_{NW}$, and OA, methods 7 and 1 ranked among the best, from which method 7 was the overall best based on the quantitative results. Additionally, the 11th WMR placed third.

We have evaluated method 7 against methods 1 and 11 using McNemar's test to investigate the improvements of the final results in both classes (Table 3). As suggested in Table 3, a significant accuracy improvement was achieved by Method 7. This improvement is more pronounced for the non-water class of method 1 and the water class of method 11 (*p*-values < 0.001).

**Table 3.** McNemar's test for method 7 against methods 1 and 11 (Table 2).

| Year | Method | | | | | | | |
|---|---|---|---|---|---|---|---|---|
| | 7 vs. 1 | | | | 7 vs. 11 | | | |
| | Water | | Non-Water | | Water | | Non-Water | |
| | $\chi^2$ | *p*-Value | $\chi^2$ | *p*-Value | $\chi^2$ | *p*-Value | $\chi^2$ | *p*-Value |
| **2018** | 6.87 | 0.01 | 8.07 | 0.01 | 46.71 | 0.001 | 7.91 | 0.01 |
| **2019** | 6.19 | 0.02 | 13.05 | 0.001 | 32.82 | 0.001 | 6.92 | 0.01 |
| **2020** | 1.78 | 0.1 | 26.46 | 0.001 | 57.46 | 0.001 | 6.34 | 0.02 |

According to Figure 4, OA performance in the SI category was the highest for method 1 (NDWI), followed by methods 5 (ANDWI), 3 (AWEI$_{sh}$), 4 (WI$_{2015}$), and 2 (MNDWI), respectively. Further, in the MI category, methods 7, 9, 8, and 6 achieved the highest to lowest OAs. According to the results, the final OA is higher when vegetation and water indices are combined (MI category). For example, the NDWI (1st method in the SI category) resulted in OAs ranging from 98.93 to 99%; however, by using NDVI and EVI vegetation indices along with NDWI (7th method in the MI category #7), the OA ranged between 99.13 and 99.24%. The Ortho$_w$ (11th) method performed the best in the TB category, followed by the LBV and Rot$_w$ approaches.

To better understand the performance of the different WMRs, Figure 5 compares the obtained results of each method in 2020 with the reference maps for Iran's central region (central plateau basin). The selected area is mainly covered by bare soils and saline regions. Only the 1st, 7th, and 11th methods, which showed greater OAs than the others, were highly similar to the reference maps (ESRI, ESA, Fmask, and JRC (median composite of monthly maps from July to October)). The majority of the regions that were falsely identified as water were saline lands, and according to in-situ surveys and local observations, there was no water in these regions due to Iran's recent droughts. Our results (Figures 4 and 5, Table 3) suggest that the 7th method outperformed other WMRs both visually and quantitatively. As a result, this article used method 7 to map SWs and their associated long-term spatiotemporal changes.

### 4.2. Long-Term Changes of SWA

In this section, the long-term SWA changes within Iran (including the Qareh-Qum Basin) were investigated (using the 7th method) for the 32-year period from 1990 to 2021 (Figure 6). There is a declining linear regression trend in SWA (Figure 6a). The SQMK test is in agreement with the linear regression as it also showed decreasing trends in SWA for both the progressive and retrograde series without intersections (Figure 6b). Moreover, a t$_{stat}$ of 12.68, and a low probability of null hypothesis occurrence (*p*-value = 0.00006) revealed by *t*-test analysis, revealed that the downward trend (with the slope of $-225.85$) was statistically significant at a 99% confidence level. Prior to 2000, Iran experienced greater SWA, yet since 2000, the SWA has decreased to less than 50% (2015) in comparison to the wettest year (1992). According to our results, over the last 14 years, 2019 was the wettest year in Iran, with an area of 7526.58 km². Our findings were compared with the median composite of JRC monthly water maps from 1990 to 2020, illustrated in Figure 6c. As can be seen, our results are highly correlated with the JRC results, confirming the declining trend for SWA in Figure 6a. *t*-test analysis also indicated a significant statistical relationship between JRC and WMR#7.

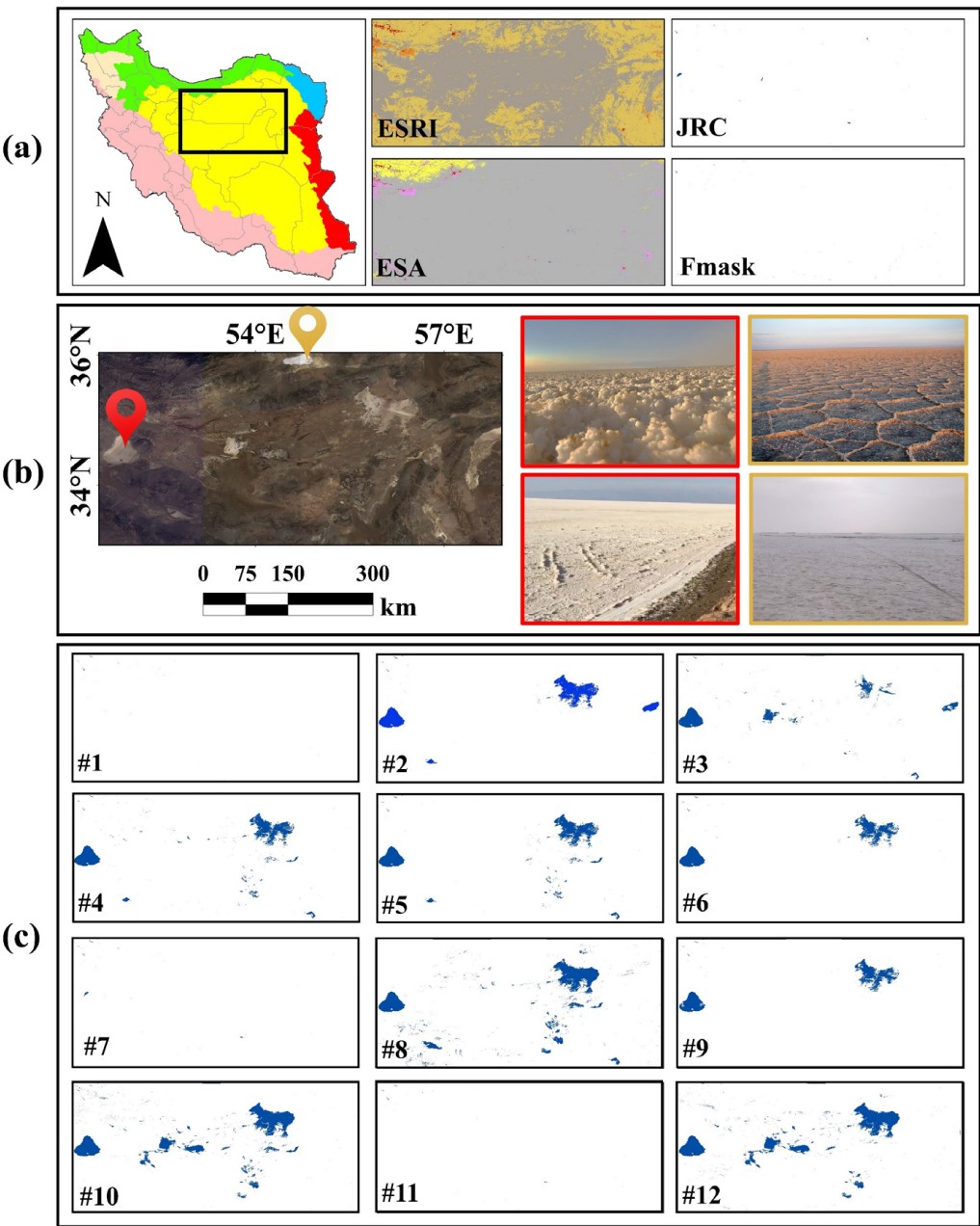

**Figure 5.** WMR comparison results for the year 2020 in the central regions (central plateau basin) of Iran: (**a**) location of the area and corresponding reference maps; (**b**) Landsat RGB composite of the study site and photos taken during field visits during July–October in 2020, (**c**) results of different WMRs (numbers are based on Table 2).

As noted in Section 2, Iran is covered by six major basins (Figure 1 and Table 1). Our analyses indicated that the Qareh-Qum Basin had an average SAW of about 10 km², and only about 9.5% of the basin is located inside Iran (which is smaller than other basins). Thus, this study did not examine the long-term SWA spatiotemporal changes in the Qareh-Qom Basin. Figure 7 displays the SWA spatiotemporal changes during 1990–2021 for the five basins that cover Iran (the Caspian Sea, the Persian Gulf, Lake Urmia, the Central Plateau, and the Easter Border), in which linear regression trends and SQMK test results can also be seen. The SQMK results were similar to the regression trends for all basins. Only the Persian Gulf Basin had an upward trend. The primary cause for the upward trend in the Persian Gulf Basin is due to the significant dam construction projects which occurred during this period to save surface water [85]. About 35% of Iranian dams are located in the

Persian Gulf Basin, most of which have the largest reservoirs nationally. In contrast, other basins experienced a downward trend that confirmed the declining trend of SWA shown in Figure 6.

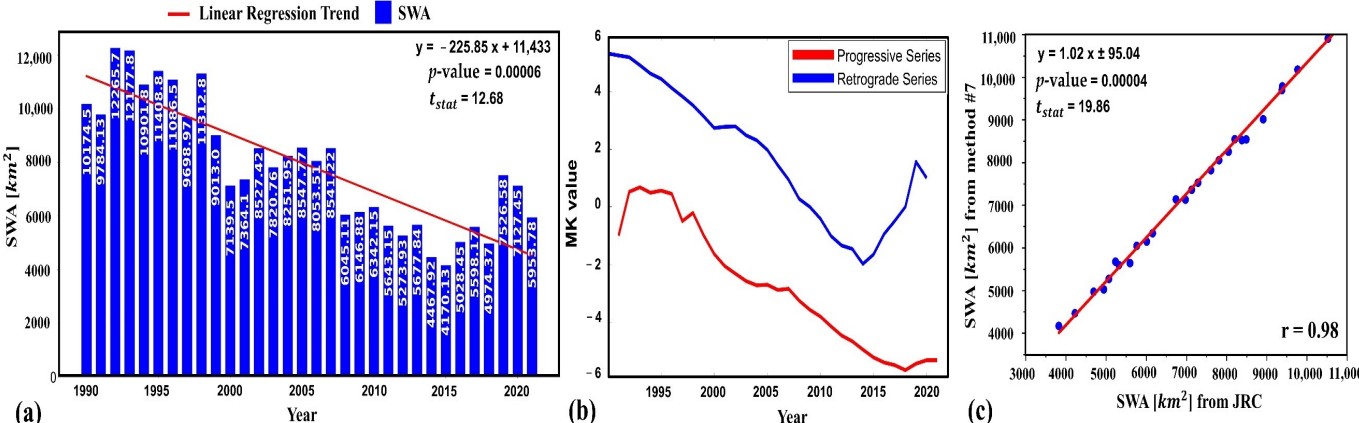

**Figure 6.** (**a**) The 32-year SWA dynamics in Iran (including the Qareh-Qum Basin), (**b**) the SQMK test results for SWA variations in Iran, (**c**) correlation analysis between results of the median composite of the JRC monthly water maps and the 7th WMR (Table 2).

Among the five major basins in Iran, the Caspian Basin had the mildest dropping slope due to it being located in the northern part of Iran next to the Caspian Sea and experiencing most of the overall precipitation (see Figure 1d). SWA has fallen sharply in the Central Plateau and Easter Border Basins as a result of severe droughts in large lakes located in the central and eastern regions of Iran, including Hamoon, Bakhtegan, and Tashk. The Urmia Lake Basin has also experienced the sharpest decline, reflecting the disappearance of water from Urmia Lake. However, restoration attempts (after 2015) have been relatively successful in restoring SWA to Urmia lake [86]. Large $t_{stat}$ values ($4.83 \leq t_{stat} \leq 11.84$) and corresponding low probabilities ($0.0007 \leq p\text{-value} \leq 0.0051$) based on $t$-test statistical analysis also suggest a rejection of the null hypothesis, implying that long-term changes in all basins are 99% confident.

The changes to some of the major lakes (Urmia, Hamoon, Bakhtegan, Maharloo, and Tashk) are shown in Figure 8 over 5-year intervals to illustrate the declining SWA trend. Lake Urmia, located in northwestern Iran (the second largest salt lake globally), is shrinking due to recent water scarcity. Lake Urmia's SWA continuously declined from 1990 to 2015, from about 6000 km$^2$ in 1995 to about 1200 km$^2$ in 2015. In comparison to 2015, Urmia's SWA grew by more than 100% in 2021 as a result of several recoveries and restoration plans. Hamoon Lake is the only lake in eastern Iran, located in the Easter Border Basin. Hamoon Lake's SWA fell from 2675.92 km$^2$ in 1990 to 65.14 km$^2$ in 2021, which accounted for a loss of more than 95% of water extent. The Bakhtegan, Maharloo, and Tashk lakes are located close to each other in the southern part of the Central Plateau Basin (Figure 8c). A sharp downward trend was also observed for all three lakes, where the current SWA in 2021 is 4.2 km$^2$, more than a 99% drop as compared to 1995.

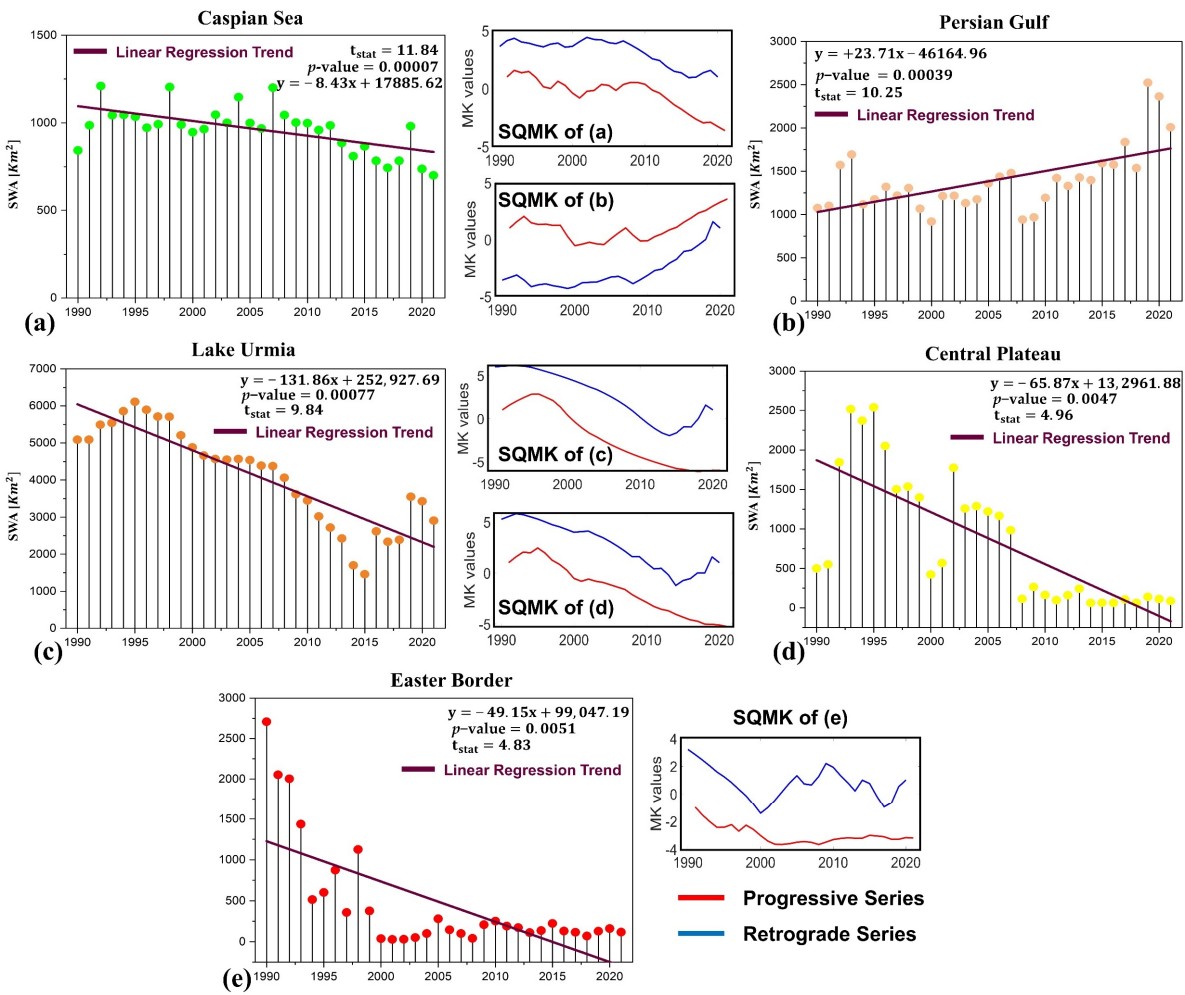

**Figure 7.** The 32-year SWA dynamics in Iran's major basins and the corresponding SQMK results: (**a**) Caspian Sea, (**b**) Persian Gulf, (**c**) Lake Urmia, (**d**) Central Plateau, and (**e**) Easter Border (since only 9.5% of the Qareh-Qum Basin is located inside Iran, this study did not examine this basin separately).

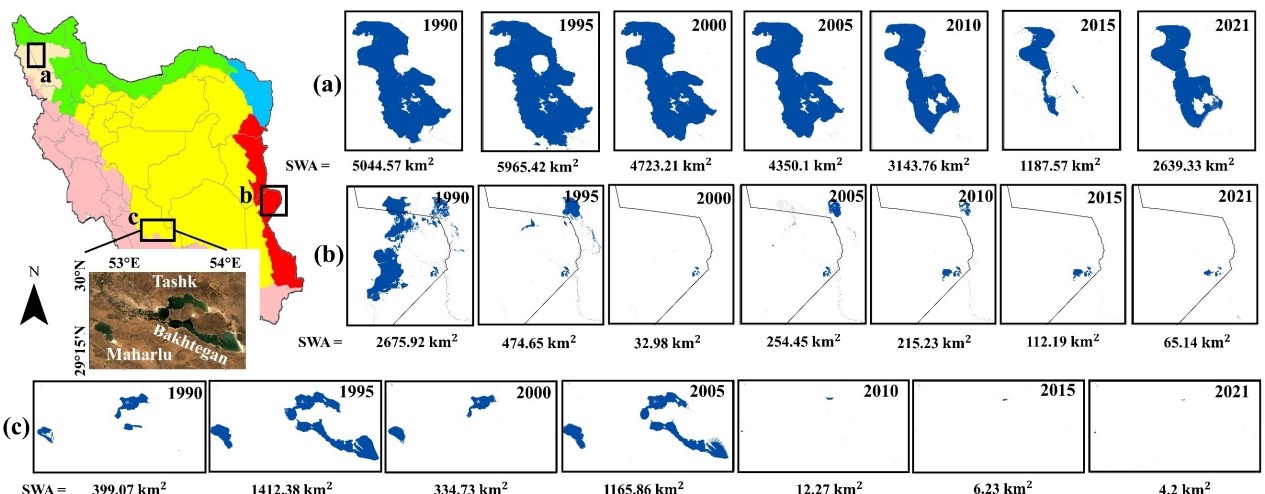

**Figure 8.** The 5-year changes of (**a**) Lake Urmia, (**b**) Hamoon lake, and (**c**) Bakhtegan, Tashk and Maharlu lakes. An RGB Landsat composite for 1995 demonstrates the location of the Bakhtegan, Tashk, and Maharlu lakes.

Due to the uneven distribution of the surface water, rainfall, topographic, and climatic conditions in Iran, the reservoirs are considered to be the primary source of freshwater supply in the country. Because of the importance of the reservoirs, changes in SWA for five of the old major reservoirs in Iran are shown in Figure 9 (Doroudzan, Kazemi, Latyan, Zayanderud, and Sefidrud). Since Iran experienced a maximum SWA in 1992 (see Figure 6), this year was used as a reference to compare the latest year in our study (2021). SWA has decreased in all five reservoirs. The Latyan, Zayanderud, and Sefidrud reservoirs have experienced steeper changes: 67.7, 68.8, and 61.8% reductions in SWA, respectively. Further, the Kazemi and Doroudzan reservoirs experienced the least change (16 and 30%) in SWA due to the complex local topography and depth of the valley they are located within.

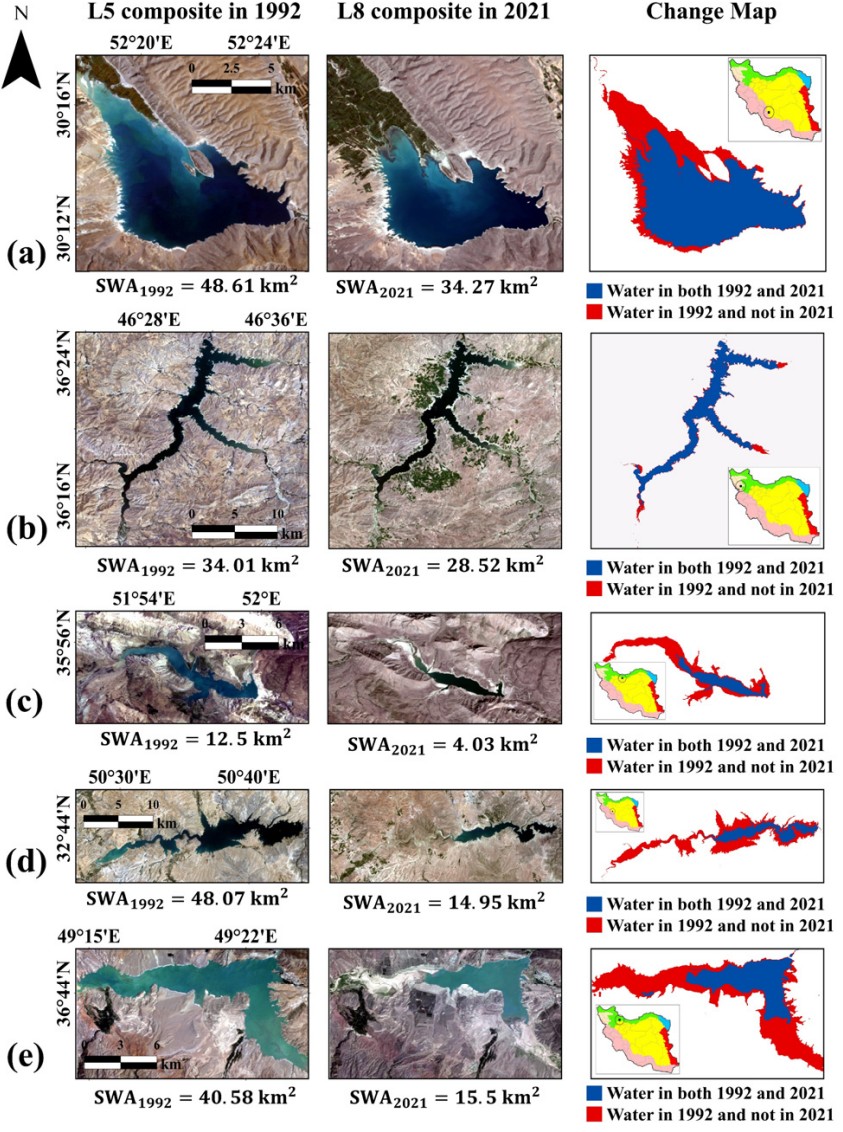

**Figure 9.** SWA changes between 1992 (wettest year in Iran (Figure 6)) and 2021 (latest status) for the (**a**) Doroudzan, (**b**) Kazemi, (**c**) Latyan, (**d**) Zayanderud, and (**e**) Sefidrud reservoirs.

*4.3. Water Frequency Map*

To better understand SWA dynamics, in Figure 10, the annual water frequency map (WFM) in Iran is shown. It shows how many times each pixel has been classified as water annually over the past 32 years. Areas with higher values are considered permanent waters without changes, while areas with smaller values are related to pixels that have changed either from non-water to water or from water to non-water. The low-frequency pixels are

mainly newly established dams or dried lakes and rivers, respectively. Figure 10a shows the Anzali lagoon, where SWA is declining, but in the same basin (Caspian Sea), modest changes in SWA can be seen within the Gorgan Gulf (Figure 10b). Figure 11c,d also represent the WFM for the Karkhe and Dez reservoirs, which were launched in 2001 and 1963 in the Persian Gulf Basin. In the Karkhe reservoir (Figure 10c), the river's main channel is of a higher frequency than the reservoir itself, yet in the Dez reservoir (Figure 10d), most areas were classed as water over the past 32 years. For the Maharlu, Bkhtegan, Tashk, Urmia, and Hamoon lakes in Figure 10e–g, SWA has declined due to the new dam construction projects. It should also be noted that the WFM is sensitive to local topography and its gradient. For instance, the Dez reservoir has a complex topo-bathy with steep slopes (i.e., deep valley), which makes it insensitive.

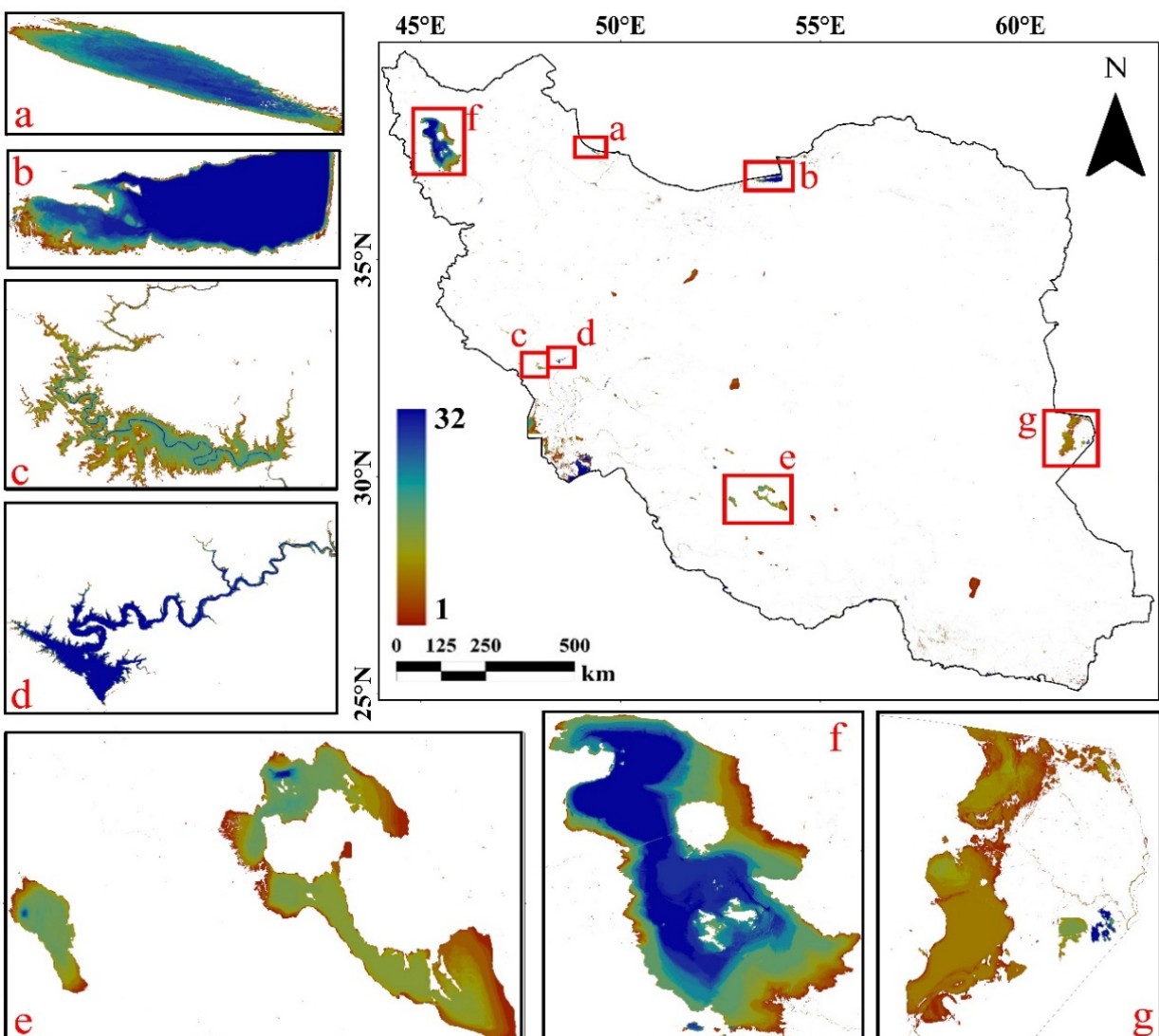

**Figure 10.** Surface water area frequency map (WFM): (**a**) Anzali lagoon, (**b**) Gorgan gulf, (**c**) Karkhe Reservoir, (**d**) Dez Reservoir, (**e**) Bakhtegan, Tashk and Maharlu lakes, (**f**) Urmia Lake, (**g**) Hamoon lake.

*4.4. Correlation with Environmental Variables*

Figure 11 compares the long-term variations in total SWA (including the Qareh-Qum Basin) with four environmental variables: annual mean precipitation rate (P), annual mean temperature (T), and mean NDVI for Iran from October to October. The results demonstrate that P and NDVI experienced an overall downward trend, which is in agreement with SWA changes for the same period in Iran. Comparatively, T showed a general rising tendency. A

decrease in P, as well as an increase in T, has aggravated drought conditions in Iran [31,87]. The *t*-test results also revealed that P, T, and NDVI slopes are of a 99% statistical confidence.

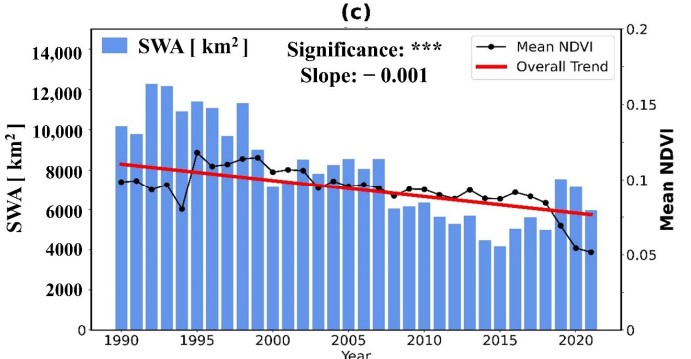

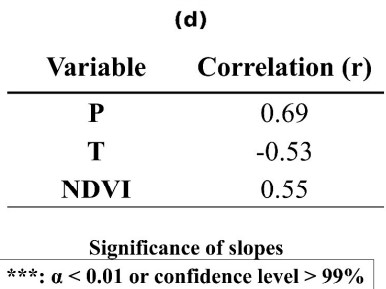

**Figure 11.** Long-term comparison of SWA against environmental variables (from FLDAS): (**a**) annual mean precipitation rate (P), (**b**) annual mean temperature (T), (**c**) annual mean NDVI, and (**d**) correlation analysis between SWA and P, T, and NDVI.

A point-by-point comparison between SWA and P had a similar pattern of ups and downs, implying a direct correlation between changes in P with changes in SWA. P also had the highest correlation with SWA changes (r = 0.69) among all other variables. For example, 2019 has seen the most rain and has had the greatest SWA for the last 15 years. Moreover, sharp drops in SWA corresponded to a significant decrease in P (e.g., 2000, 2008, 2013, and 2021). Overall, lower temperatures have been recorded in years with more SWA. In contrast, T tends to be higher in years with less SWA. The correlation analysis indicated an inverse relationship between T and SWA changes. NDVI changes mostly had a similar point-by-point trend when compared with SWA, showing the highest correlation following that of P with SWA variations (r = 0.55). The decreasing trend of NDVI might also indicate the rising severity of drought in Iran and a decline in crop acreage that can threaten food security.

## 5. Discussion

### 5.1. WMRs

We first tested twelve WMRs to select the most accurate one so as to detect spatiotemporal changes in SWA using Landsat imagery over Iran. Initially, the separation of the target classes (water and non-water) was examined using two criteria: JM and TD, in the feature space used for each technique (Figure 4a). Among the twelve WMRs, methods 1, 7, and 11 provided a higher separation (between the target classes) capability than all other methods. These methods (WMRs 1, 7, and 11) also ranked in the top three based on OA, water UA, and non-water UA, indicating that higher separable feature spaces can increase detection accuracy (Figure 4b). In other words, having a high-separable feature space allows target classes to be more accurately classified by a simple threshold. Consequently, using separability measures and accuracy parameters can be used as promising tools to compare the performances of various WMRs.

Selected WMRs were related to three categories: SI, MI, and TB. The feature spaces with a combination of water and vegetation indices (MI methods) resulted in higher separability between the target classes than those with only water indices (SI approaches) and could also achieve higher OAs and UAs than other SI methods. Thus, using vegetation and water indices together can improve water identification in remote sensing satellite images for arid environments similar to Iran [2]. Furthermore, MI methods also outperformed TB methods regarding separability measures, OAs, and UAs. The best method for SW detection was the 7th method (Table 2), noted as an MI technique.

In general, most of WMRs were able to detect SWs ($UA_w$ in Figure 4b), but their ability to identify non-water classes was a determiner of the best method for surface water mapping ($UA_{nw}$ in Figure 4b). The reason most WMRs yielded a lower $UA_{nw}$ was that the salt marshes in arid regions were misclassified as water (Figure 5). The NIR band reflects higher energies than the SWIR bands in these areas [88], meaning that NIR bands can be more discriminative than those of SWIR. Figure A1 (in Appendix A) also illustrates the spectral signatures of several land cover types and their averages concerning the spectral signature of water (collected from the USGS Spectral Library) [71]. Because of the higher reflectance in the NIR spectral band, the target classes are better distinguished by this than by the SWIR spectral band. As a result, NIR-based methods outperformed SWIR-based ones in SW mapping. For example, WMR #1 (based on NDWI using G and NIR bands) achieved greater accuracy than WMR #2 (based on MNDWI using G and SWIR bands).

Additionally, methods using NIR and SWIR together also performed better than SWIR-only techniques. For instance, $WI_{2015}$, $AWEI_{sh}$, and ANDWI outperformed MNDWI. TB category methods also used SWIR bands, which led to a lower performance than the SI and MI approach. It should be highlighted that the calculation of the TC coefficients by orthogonalization techniques ($Ortho_w$) achieved a higher classification accuracy than the rotation-based methods ($Rot_w$). Both $Ortho_w$ and $Rot_w$ used TOA reflectance images to calculate the TC coefficient, which might be another reason for their lower performance since other studies have concluded that using SR images can be more effective than TOA images for different applications [63].

*5.2. SWA in Iran*

Iran's long-term spatiotemporal SWA variations exhibited a general downward trend (Figure 6), which was evident in the well-known lakes and dam reservoirs (Figures 8 and 9). JRM reference water mapping also confirmed the declining trend of SWA in Iran (Figure 6). Examining SWA changes in Iran's major water basins revealed that only the Persian Gulf Basin experienced an overall increasing trend due to massive dam construction projects that store SW (Figure 7) [85]. Among all five of the investigated basins in this study, the Lake Urmia Basin had the steepest drop slope, where its SWA in 2015 shrunk to less than 60% of that measured in 1996. It was after 2015 that the restoration projects for lake Urmia became relatively successful and resulted in a greater SWA [86]. The Caspian Basin has experienced a milder descending slope for SWA shrinkage, mainly due to the topographic and climatic characteristics of the basin, which is located in Iran's rainiest region [89]. Declining SWA in the Central Plateau and Easter Border basins occurred mainly due to the dryness of the main lakes in the region, including Hamoon Lake, Bakhtegan Lake, Tashk Lake, and Maharloo Lake.

The precipitation rate showed a similar declining pattern concerning changes in SWA (Figure 11). In contrast, the temperature has experienced an overall increasing trend, negatively affecting SWA. A growing trend in temperature can increase potential evaporation and the atmospheric vapor pressure deficit, accelerating the actual evaporation of SWs [90]. Precipitation was the most correlated variable with variations in SWA. Moreover, the decreasing trend in precipitation and the increasing trend in temperature exacerbated drought in Iran (from 2000) [87,91]. Additionally, the declining trend of NDVI can be a result of crop acreage decrease and lower agricultural product performance, which confirms water scarcity in the agricultural sector (as the largest consumer of freshwater) [92,93].

As for the uneven distribution of surface water resources throughout the country due to climatic conditions and terrain topography (Figure 7), there are not enough water resources to meet the demands for industrial, agricultural, and urban-related uses in some regions. To maintain a balance between water demand and water supply, dam construction has been considered one of the leading solutions in Iran, especially within the Persian Gulf Basin [85]. Another solution to overcome this shortage is to transfer water from water-rich basins such as the Persian Gulf to the central regions in the Central Plateau Basin [94]. In recent years, water transfer from the surrounding seas and desalination processes have also been proposed for the future as alternatives in this field [95].

### 5.3. Flood and Drought Events

Among the environmental factors, precipitation was the most correlated with changes in SWA (Figure 11d). Therefore, flood and drought events (as extreme hydrological events) directly affect surface water resources [96,97]. As illustrated in Figure 6, the highest SWA in Iran occurred from 1990 to 2000; this period was also reported to have the greatest number of flood incidents [98]. A downward trend in the number of flood events can be seen after 2000, similar to SWA trends [98]. Destructive and massive floods have been reported in Iran in 1992 and 2019, with SWA (Figure 6) experiencing greater values. Also, hydrological droughts occurred right after 2000, when SWA declined by one-third during the wettest year and resulted in socioeconomic droughts [99,100]. For example, SWA showed a drastic decline during 1999–2000. There have been severe droughts in Iran during this period [101]. A severe drought was also reported during 2008–2009, which led to a dramatic drop in SWA from about 8500 to 6000 (a 30% decrease) [89]. It should be highlighted that natural and environmental variabilities and climate change are not the only reasons for the recent water scarcity in Iran. Several other factors have to be considered, such as rapid population growth, migration and urbanization, inefficient agriculture, the dream of self-sufficiency in food, cheap water and energy, deep wells, etc. [102].

### 5.4. Uncertainties and Future Trends

In this study, 30-m Landsat satellite images were used to map SW. High-resolution satellite images, such as those produced by Sentinel-2 (10 m), can prevent the possible confusion caused by mixed pixels, especially at the borders of water bodies, and can also help identify narrow rivers [21]. Additionally, the synergistic use of optical images with Sentinel-1 synthetic aperture radar (SAR) images or polarimetric SAR data can also improve SW detection [20,103]. SAR-based images can be used even when the sky is cloudy, allowing researchers to study spatiotemporal changes more densely. Moreover, this study only analyzed the intra-annual changes in SWA during the warmest months (July–October for each year). Inter-annual analysis, which considers other periods within each year, can provide a better understanding of the long-term change trends, seasonal fluctuations, and those prone to flood and drought events [11,34]. However, generating a free-from-gap image composite over Iran for other periods is challenging. Since the compared WMRs do not require ground-truth data and are applied to any Landsat scene, their application would increase the temporal frequency and improve the performance of global reference maps such as JRC.

It should be highlighted that this article used a constant threshold (mainly the default threshold 0) for each WMR. The best threshold for surface water extraction can be found using Otsu's thresholding, which has improved the final results for some scholars [104]. However, other articles have reported less accurate results (using Otsu's thresholding method) than those measured using the default threshold [105]. Therefore, it is important to investigate the Otsu method first to see if it enhances classification accuracy.

The methods used in this study were pixel-based classification approaches and can only identify surface waters. Therefore, they were incapable of detecting water pixels below the vegetation canopy [31]. Methods that integrate image segmentation with pixel-based results can boost classification accuracy by taking both spectral and spatial information

into account [106,107]. Probable water existence in the hill-shaded areas can be considered as another uncertainty. This article used unsupervised classification techniques to map SW. Future research can focus on developing novel automatic methods for preparing training data for supervised classifiers, allowing the comparison of different machine-learning-based models such as SVM, RF, DNNs, and ensemble classifiers [32,108].

This study was designed and implemented according to the capabilities of the Google Earth Engine (GEE) platform. Although GEE provides online data processing without the need to download, it has limited built-in algorithms, which reduces the possibility of employing a wide variety of models. Moreover, GEEs memory capacity and run-time constraints have to be considered in different studies and applications [7,62].

## 6. Conclusions

Surface waters are the main source of freshwater supply in arid and semi-arid regions, such as those found in Iran. By using about 18,000 Landsat satellite images, the spatiotemporal variations in Iran's surface water area were examined using the Google Earth Engine cloud processing platform from 1990 to 2021. The following items are the main conclusions of this research:

- Preliminary results revealed that, from the twelve WMRs (of different water mapping rules of SI, MI, and TB), those providing a higher separation between the two target classes (water and non-water) lead to higher overall classification accuracy;
- The results also indicate that methods using the NIR band can achieve higher accuracy than those using only SWIR or in combination with NIR with SWIR (NIR + SWIR) bands. Among the twelve WMRs from this study, the MI-based method WMR #7 (EVI < 0.1 and (NDWI > NDVI or NDWI > EVI)) was selected as the most accurate approach to surface water mapping;
- Of the five major basins that cover Iran, only the Persian Gulf Basin had an upward trend for SWA. In contrast, other basins experienced a downward trend in SWA;
- There was a declining trend for total SWA from 1990 to 2021 due to drought. Prior to 2000, Iran experienced higher SWA values, but since 2000, SWA in 2015 has declined to less than 50% when compared to the wettest year (1992);
- An analysis of the environmental variables through the same period (1990–2021) also confirmed overall SWA trends. Precipitation (P) and NDVI experienced an overall downward trend (direct correlation with SWA), but temperature (T) showed a general rising tendency (inverse correlation).

The current research results show the plausibility of remote sensing methods to detect and monitor long-term changes in surface water area in arid and semi-arid regions (e.g., Iran). A comparison of the results with hydrological variables and other observations supports the theory of ongoing surface water scarcity in Iran. This study framework can be used to assess long-term freshwater challenges due to climatic changes in arid regions.

**Author Contributions:** Conceptualization, A.T.D., M.J.V.Z., M.J. and A.M.; methodology, A.T.D., H.G., M.J. and A.M.; developing codes and software, A.T.D. and H.G.; supervision, M.J.V.Z. and A.M.; writing—original draft preparation, A.T.D., H.G. and M.J.; writing—review and editing, M.J.V.Z., M.J. and A.M. All authors have read and agreed to the published version of the manuscript.

**Funding:** This research received no external funding.

**Data Availability Statement:** Not applicable.

**Acknowledgments:** The authors sincerely appreciate NASA and USGS for supporting the Landsat program, which provides valuable earth-observed data for researchers and scientists worldwide. The authors would also like to express their gratitude to the GEE team for providing an online cloud processing platform with petabytes of remote sensing data.

**Conflicts of Interest:** The authors declare no conflict of interest.

## Abbreviations

| | |
|---|---|
| **AWEI** | Automatic Water Extraction Index (for shaded images with dark surfaces (AWEIsh) and shadowless images (AWEInsh)) |
| **DEM** | Digital Elevation Model |
| **DNN** | Deep Neural Network |
| **ESRI** | Environmental Systems Research Institute |
| **EVI** | Enhanced Vegetation Index |
| **FLDAS** | Famine Early Warning Systems Network Land Data Assimilation System |
| **GEE** | Google Earth Engine |
| **GRD** | Ground Range Detected |
| **LBV** | L: general radiance level, B: visible-infrared radiation balance, V: radiance variation vector |
| **MI** | Multi-Index |
| **MODIS** | Moderate Resolution Imaging Spectroradiometer |
| **NDWI** | Normalized Difference Water Index |
| **OLI** | OLI Operational Land Imager |
| **P** | Annual mean precipitation rate |
| **QA** | Quality Attribute map |
| **Rotw** | Wetness component of TC transformation derived by transforming Principal Components (PC)-based rotated axes |
| **RS** | Remote Sensing |
| **SC** | Supervised Classification |
| **splib07b** | Spectral library version 7 |
| **SQMK** | Sequential Mann–Kendall |
| **SVM** | Support Vector Machine |
| **SW** | Inland Surface Waters |
| **TC** | Tasseled-Cap transformation |
| **TD** | Transformed Divergence |
| **TM** | Thematic Mapper |
| **UA** | User Accuracy |
| **USGS** | United States Geological Survey |
| **WMR** | Water Mapping Rule |
| **WI2006, WI2015** | Water Index |
| **ANDWI** | Augmented NDWI |
| **API** | Application Programming Interface |
| **AVHRR** | Advanced Very High-Resolution Radiometer |
| **ESA** | European Space Agency |
| **ETM+** | Enhanced Thematic Mapper Plus |
| **Fmask** | Fmask Function of the mask |
| **GHSL** | Global Human Settlement Layer |
| **JM** | Jefferies-Matusita |
| **JRC** | Joint Research Center |
| **L5, L7, L8** | Landsat 5, 7, 8 |
| **LSWI** | Land Surface Water Index |
| **ME** | Middle East |
| **MNDWI** | Modified NDWI |
| **NASA** | National Aeronautics and Space Administration |
| **NDVI** | Normalized Difference Vegetation Index |
| **NIR** | Near Infrared |
| **NOAA** | National Oceanic and Atmospheric Administration |
| **OA** | Overall Accuracy |
| **Orthow** | Wetness component of TC transformation derived by orthogonalization techniques such as Gram-Schmidt |
| **RB** | Rule-Based |
| **RF** | Random Forest |
| **SI** | Single-Index |
| **SLC** | Scan Line Corrector |

| | |
|---|---|
| **SRTM** | Shuttle Radar Topographic Mission |
| **SWA** | Surface Water Area |
| **SWIR** | Short-Wave Infra-Red |
| **T** | Annual mean temperature |
| **TB** | Transformation-Based |
| **WFM** | Water Frequency map |

## Appendix A

Figure A1a displays the spectral signature of the various landcover classes extracted from the splib07. These landcover classes can be found in Iran. The splib07 includes hundreds of samples from a broad range of categories. Each signature in Figure A1a represents a specific category. In Figure A1b, the mean spectral signature of the non-water class is compared to the water class.

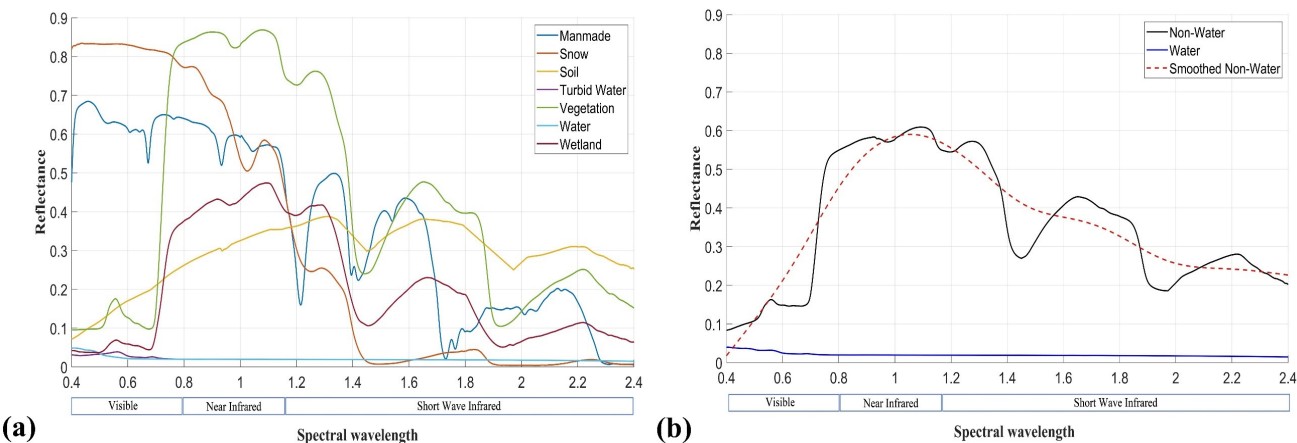

**Figure A1.** Spectral signature of (**a**) different landcover classes and (**b**) water and non-water (derived from USGS spectral library (splib07)).

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
