# Peer review of "Monitoring Long-Term Spatiotemporal Changes in Iran Surface Waters Using Landsat Imagery"

_remotesensing, doi:10.3390/rs14184491_

Round 1

Reviewer 1 Report

11 – who is from Department of Civil and Environmental Engineering, San Jose State University, CA 95192, USA – no. 4?

35 - 136  - There is no description of available databases with changes of surface waters. For example: Jean-Francois Pekel, Andrew Cottam, Noel Gorelick, Alan S. Belward, High-resolution mapping of global surface water and its long-term changes. Nature 540, 418-422 (2016). (doi: 10.1038 / nature20584)  - JRC database. Why is it necessary to look for new methods if we have a global database with a resolution of 30m? What is the purpose of this study?

207  - How have build-up areas created after 2016 been removed? (GHSL contains data up to 2016)

213 - What if the water was in hill-shaded areas and was removed?

230  - Why was NDVI computed from NOAA AVHRR used if You already computed NDVI from Landsat?

269 - Line 204 says that images from Scan Line Corrector (SLC) -related errors have not been used

279 - How was the shaded areas mask created? What rule.

402 – what value of correlation?

457 - what do the colours on the map mean?

617 – 647   - where were the biggest mistakes? which objects were characterized by false positive and false negative errors? Are your results better than the JRC?

Reviewer 2 Report

This is a brilliant piece of manuscript. I enjoyed reading it and vote for its acceptance for the publication.  

Author Response

We do appreciate reviewer #2 consideration.

Reviewer 3 Report

The manuscript titled ‘Monitoring Long-term Spatiotemporal Changes of Iran Surface Waters Using Landsat Imagery’ examined the twelve water mapping rules (WMRs) using Landsat 4, 5, and 7 satellite images for surface water (SW) classification. After that, the variations of surface water were examined in Iran from 1990 to 2021 using the optimal classification method. They found that near-infrared (NIR)-based methods obtain good results in arid regions and SW in Iran has an overall downhill trend. The influence factors of temperature presented an upward trend, precipitation and NDVI experienced a downward trend, and precipitation showed the highest correlation with SW area changes. However, by closely examining the manuscript, I found the research background and results are not solid. Firstly, as you claimed in Section 3.1.7, there are global water/no water datasets that cover a long period (since 1984). So I wonder to know the motivation of this study to produce a new surface water area product. Secondly, the Introduction presented many details about the classification methods, but readers have trouble in understanding the problems in the current classification methods or existing products, please adjust the Introduction and form a good storyline. Thirdly, the optimal classification method used in this study has produced the 32-year SWA dynamics in Iran, but the comparison result with JRC indicated a significant difference in Fig.6c. So I doubt the performance of the optimal method obtained in this study, more comparisons with existing products or official data should be conducted. In addition, surface water (SW) and surface water area (SWA) are both included in this study, but readers can only observe the extraction or variations of SWA, rather than SW. In my opinion, SW has many aspects, i.e. surface soil moisture, surface water area, etc., so the term should be unique. So considering the innovation of this study, motivation of research, and reliability of results, I think this manuscript is unsuitable for publication in Remote Sensing.

Reviewer 4 Report

This manuscript compares different common methods of mapping surface water using Landsat imagery across Iran. The authors provide a comprehensive comparison of methods with valuable recommendations for mapping surface water in environments similar to Iran. In particular, the use of NIR in water mapping rules (WMR) performed better than SWIR-based methods in Iran. The application of the best WMR to assess trends in surface water also provided useful insight. I recommend this manuscript is suitable for publication following consideration of the following comments, and correction to the minor grammatical errors (listed below).

Line 117 – change ‘to be well-performed’ to ‘to be a well-performed’

Line 128 – should be ‘that SW here refers to Iran’s inland SW …’

Line 181 – 184 – I thought that reference [45] (Q. Liu, G. Liu, C. Huang, S. Liu, and J. Zhao, "A tasseled cap transformation for Landsat 8 OLI TOA reflectance images," in 798 2014 IEEE Geoscience and Remote Sensing Symposium, 2014: IEEE, pp. 541-544.) talks about Landsat 8 TCT parameters? What did you use for Landsat 5 and 7? Can you please what you used in the text?

Figure 2 caption – change to ‘number of images used in each year.’

Table 2 – It could be useful for include formulas here (it could replace the ‘Required bands’ column).

Line 355 – I would mention that the 7th method is from Table 2.

Paragraph 367-378 – I suggest adding the WMR name next to the method number would improve the readability of this paragraph (like you did in paragraph 386-394).

Line 399 – change to ‘Figure 5 compares’

Figures are supposed to be mentioned in the text before the actual figure.

Line 542 – ‘Thus, using vegetation and water indices together can improve water identification in remote sensing satellite images’. I suggest adding that this is for environments similar to Iran, since other studies have found other indices were better in their study area.

Line 624 – change to ‘spatiotemporal changes more densely’.

Minor grammatical errors:

Line 11 – No author is associated with the 4th address

Line 57 – change ‘common’ to ‘commonly’

Figure 3 – typing error ( ‘,,’) in the box starting with ‘(NDWI,’

Line 310 – typing error ‘))’

Line 318 – typing error ‘maps reference maps’

Line 448 – remove double brackets around P value

Line 490 – change ‘nonwatery’ to ‘nonwater’

Reviewer 5 Report

The authors used the Rule-Based algorithms to analyze the satellite images collected during 1990 to 2020 to derive the surface water areas. They evaluated various algorithms according to their performance measured with Transformed Divergence and Jefferies-Matusita distances. However, the authors need to separate the change of surface water areas due to artificial processes like building dams from the natural change. When the authors try to link the decadal change of SWA with the environment change like precipitation change, they also need to consider the change of evaporation.
